# Influence of Membrane Fouling and Reverse Salt Flux on Membrane Impedance of Forward Osmosis Microbial Fuel Cell

**DOI:** 10.3390/membranes12111165

**Published:** 2022-11-19

**Authors:** Yang Zhao, Liang Duan, Xiang Liu, Yonghui Song

**Affiliations:** 1Chinese Research Academy of Environmental Sciences, Beijing 100012, China; 2State Key Joint Laboratory of Environment Simulation and Pollution Control, School of Environment, Tsinghua University, Beijing 100084, China

**Keywords:** forward osmosis membrane, membrane fouling, reverse salt flux, concentration polarization, mass transfer model

## Abstract

The forward osmosis membrane (FO membrane) is an emerging wastewater treatment technology in bioelectricity generation, organic substrate removal and wastewater reclamation. Compared with traditional membrane materials, the FO membrane has a more uniform water content distribution and internal solution concentration distribution. In the past, it was believed that one of the important factors restricting power generation was membrane fouling. This study innovatively constructed a mass transfer model of a fouling membrane. Through the analysis of the hydraulic resistance coefficient and the salt mass transfer resistance coefficient, the driving force and the tendency of reverse salt flux during membrane fouling were determined by the model. A surprising discovery was that the fouling membrane can also achieve efficient power generation. The results showed that the hydraulic resistance coefficient of the fouling membrane increased to 4.97 times the initial value, while the salt mass transfer resistance coefficient did not change significantly. Meanwhile, membrane fouling caused concentration polarization in the FO membrane, which enhanced the reverse trend of salt, and the enhancement effect was significantly higher than the impact of the water flux decline caused by membrane pollution. This will make an important contribution to research on FO membrane technology as sustainable membrane technology in wastewater treatment.

## 1. Introduction

Compared with traditional separation materials, the forward osmosis (FO) membrane has a more uniform water content distribution and internal solution concentration distribution [1]. During operation, the reverse salt flux driven by the draw solution will improve the conductivity of the anolyte, reduce the solution impedance and also reduce the FO film impedance. Therefore, the FO film has a lower membrane impedance, which enhances the power generation performance of an osmosis microbial fuel cell (OsMFC) [2].

Although membrane fouling in an FO process is generally less severe than that in pressure-driven membrane processes, it is still an inevitable phenomenon in an osmosis bioreactor and can lead to additional resistance that reduces the osmotic pressure and water flux, thereby increasing the capital and operational costs. When treating wastewater, organic fouling due to the presence of organic matter and microorganisms can become significant, and in this aspect, a coupled osmosis bioreactor may be more advantageous over an integrated bioreactor, because the bioreactor unit in the coupled osmosis bioreactor acts as a pre-treatment step and reduces organic contents in the liquid stream before it enters the FO unit. When fouling occurs, backwash is a commonly used method to remove foulants and restore the FO performance. More severe fouling would require chemical cleaning, either in situ or ex situ. Regardless of the cleaning method, the microbial community in the anode would be affected, for example, by a shock from a high-salinity solution or chemical toxicity. Fouling alters the membrane properties, and in some cases, it may benefit energy recovery in an osmosis bioreactor. Considering that membrane pollution and concentration polarization are inevitable under long-term operating conditions, the difference between the concentration of salt on the membrane surface and that of the extract solution is reduced [3]. However, Gu analyzed the experimental data in the process of studying sodium alginate CaCl_2_ membrane pollution and found that the reverse salt flux of the FO membrane was increased to a certain extent, and the salinity of the feed solution also increased significantly under the influence of this factor, which was significantly different from the group without membrane pollution [4]. Zhang ‘s research on this type of battery found that the diffusion flux of the FO membrane to Na^+^ and Cl^−^ in the solution increased after pollution, while the increase in reverse salt flux and ion flux directly affected the change in membrane impedance [5]. In the early stage, the pollution of an ion-exchange membrane in MFC will lead to a decline in power generation performance, but there is still no research on the OsMFC system. It was found that the current generation of an OsMFC was increased by 34% upon membrane fouling without water flux. This is because the fouled FO membrane showed a significantly higher flux of protons and other ions than the pristine membrane, resulting in lower internal resistance and higher current generation. In order to explore the change in membrane impedance after membrane pollution, it is necessary to deeply study the mechanism of FO membrane pollution effects on reverse salt flux, and the changes in ion flux and membrane impedance caused by reverse salt flux were analyzed.

In this paper, the power generation performance and internal resistance of OsMFC are analyzed in detail, and the trends are analyzed. The results show that the FO membrane has better performance after pollution. Further, a permeability resistance model after FO membrane fouling was established, and the model parameters were optimized and improved based on the experimental results so as to improve the accuracy and application value of the model. The driving force and resistance of salt backflow during membrane fouling were determined, and the mechanism of membrane impedance change due to membrane fouling and increased FO salt backflow was deeply investigated.

## 2. Materials and Methods

In this study, a double-chamber OsMFC was used. The new membrane and the contaminated membrane were each used as the separation material, and the active layer was facing toward the anolyte/catholyte. The water flux was measured based on the volume change of the catholyte. For the operation of OsMFC, artificial wastewater was selected as the anolyte, and 1.5 M NaCl was used as the draw solution. OsMFC was run for 5 cycles in total. When the current was lower than 0.1 mA, the membrane and electrolyte were changed according to the experimental scheme, and the experimental measurement was conducted again [6]. The new membrane and the contaminated membrane were operated alternately in the same MFC so that the interference of other factors could be eliminated. All experiments were conducted at room temperature.

### 2.1. Ion Diffusion Experiment

An abiotic system was used to measure ion flux. The measuring device was a dual-chamber MFC with a volume of 250 mL. The anode and cathode were made from carbon paper and carbon felt, respectively, and the effective area was 2 cm^2^. The membranes used in the test were fresh and fouled FO membranes, and the effective area of the membrane was 3.8 cm^2^. The set anolyte and catholyte were 10 mM KNO_3_ and 1 M NaCl, respectively. All reagents were from Sinopharm (analytically pure). The salt solution was prepared with deionized water (Millipore Inc., Burlington, MA, USA), and the chamber was magnetically stirred [7]. During sewage treatment, a constant current of 0.05 mA·cm^−2^ was applied to the test system to simulate the effect of the MFC electric field, and the temperature of all experiments was controlled at 25 °C.

### 2.2. Calculation of Reverse Salt Flux

The determination of the change in conductivity of the feed solution over time and the relationship between conductivity and solution concentration was performed using the following equation:(1)$$K=\Lambda_{m}\times C\times 10^{-3}$$
where K is the conductivity of the solution, μs·cm^−1^; $\Lambda_{m}$ is the molar conductivity, S cm^2^·mol^−1^. The correlation between molar and equivalent conductivities is $\Lambda_{m}\;$= λM, where λ is the equivalent conductivity, S cm^2^·g^−1^. M is the molar mass of the electrolyte, g·mol^−1^. During calculation and analysis, the equivalent conductivity of the solute is determined by looking up values in the table, and the solute concentration can be determined by combining the measurement results with analysis [8].

### 2.3. Proton Flux Calculation

The proton flux of the FO membrane is mainly determined based on the difference between the theoretical and measured pH values. According to the theoretical analysis, one proton in the hydrolysis system corresponds to one electron transfer, so the theoretical pH value of the catholyte can be calculated by the following expression:(2)$${\mathrm{pH}}_{\mathrm{Theoretical}}=14+\log\frac{\frac{Q}{F}-10^{-{\mathrm{pH}}_{0}}V_{d0}}{V_{\mathrm{dt}}}$$

The total transferred charge (Q) can be determined by the time integration of the current, and pH_0_ is the initial pH value of the cathode solution. This deviation of pH is mainly caused by a certain loss during proton diffusion. In this way, the actual proton diffusion flux, J_H_, can be calculated as follows [9]:(3)$$J_{H}=\frac{10^{-{\mathrm{pH}}_{\mathrm{test}}}V_{t}-10^{-{\mathrm{pH}}_{\mathrm{Theoretical}}}V_{0}}{A_{m\;}t}$$

### 2.4. SMP and EPS Analyses

A total of 50 mL of the mixed anode solution was centrifuged at 4 °C and 12,000× *g* for 15 min, and the supernatant was passed through 0.45 μ. The SMP sample to be tested was obtained after filtering with a PVDF membrane of m. EPS was extracted by the resin method. CER (Dowex^®^Marathon^®^C, Na+form, Sigma-Aldrich, Bellefonte, PA, USA) was added to a 50 mL sample according to a dosage of 70 g CER/g VSS and vibrated at 4 °C and 600 rpm for 2 h. The mixture was removed and centrifuged at 4 °C and 12,000× *g* for 15 min, and 0.45 μ was used for the centrifuged supernatant. The EPS sample containing SMP was obtained after filtering with the PVDF membrane. The amount of EPS can be obtained by subtracting the amount of SMP from the total EPS content. The Folin phenol reagent method and anthrone sulfuric acid method were used to determine the protein and polysaccharide contents in SMP and EPS. The standard samples were bovine serum protein and glucose.

The extraction method of the biofilm EPS on the filler surface is as follows: Remove the sample from the anode solution and put it into a 50 mL centrifuge tube, add 1.0 mM phosphate buffer solution (PBS), vortex for 5 min to make the biofilm fall off, continue to clean the filler surface with PBS and adjust the volume to a certain volume. Then, extract EPS with the resin method and centrifuge at 4 °C and 12,000× *g* for 15 min. Finally, use 0.45 μM membrane filtration to obtain the EPS solution to be tested [10].

The extraction method of biofilm EPS on the membrane surface is as follows: First, use physical methods to make the biofilms on most of the electrodes fall off, then use a sterile cotton swab to scrape off a small amount of residual biofilm on the membrane surface, adjust the volume with 1.0 mM PBS solution and then use the resin method to extract EPS. Determine the contents of proteins and polysaccharides in the extract and subtract the content of SMP to obtain the corresponding EPS component content. The salt mass transfer resistance of the system is not significantly increased by membrane fouling, thus enhancing the driving force of the reverse salt flux in a disguised way, causing the reverse salt flux to increase to 2.3 times that of the new membrane, which is far higher than the impact of the decline in water flux caused by membrane fouling, thus increasing the concentration on the feed liquid side, reducing the membrane resistance and overall impedance and promoting the improvement of power generation efficiency [11].

In order to distinguish loosely bound EPS (LB-EPS) and tightly bound EPS (TB-EPS), different SMP and EPS extraction methods were adopted. First, the sludge mixture was centrifuged at 6000× *g* for 5 min and then passed through 0.45 μ. SMP was obtained by membrane filtration, and LB-EPS and TB-EPS were extracted by an improved thermal extraction method.

## 3. Results and Discussion

### 3.1. Water Flux with Long-Term Operation of OsMFC

Since the operation of OsMFC involves anaerobic digestion by microorganisms and FO membrane filtration, it is necessary to study the characteristics of the FO membrane process and make appropriate improvements in order to effectively improve the overall performance of the system. In this study, the appropriate operating parameters of the reactor were selected in the research process so as to improve the state of the FO membrane and extend its service life, providing support for improving the output performance of the system. In this analysis, the change trend of membrane water flux during OsMFC treatment was analyzed under the condition of changing the concentration of the absorption solution. The relevant test data were processed, and the results are shown in Figure 1. Under the condition that the circulating flow rate remained constant at 1 L·min^−1^ with the same concentration of each extract, the membrane was run for 3–4 cycles, and the membrane was properly cleaned to ensure that its flux met the requirements. When the membrane flux is lower than the set value, it is less than 2 LMH.

From the results in Figure 1, it can be found that there are differences in the initial flux of the corresponding FO membrane under different conditions of the draw solution, and there is also a certain positive correlation between the initial flux and the concentration of the catholyte. According to the experimental results, the initial flux is 6 LMH when the concentration of the extracted solution is 0.5 M. When the former is 1 M, the latter increases to 10 LMH. As it continues to increase to 1.5 M, the initial flux rises to 13 LMH, with an increase of 117%. At the same time, it can be seen in the figure that there are differences in the operating cycle under the conditions of different concentrations of the draw, and the operating cycle changes with the opposite trend under the conditions of an increased concentration of the draw. According to the analysis, when the concentration of the draw solution is 0.5 M, the operation cycle is 20 h; when the concentration of the draw is 1 M, it is reduced to 13 h; and when the concentration of the draw is 1.5 M, it is reduced to 2 h. The analysis shows that the duration of each cycle decreases continuously after the operation cycle increases in this process. When the concentration of the extract changes, the corresponding membrane flux decreases differently, and there is a positive correlation between them, in general. It can be concluded that the initial flux of the FO membrane is increased to a certain extent after the concentration of the extract is increased, but the corresponding membrane pollution becomes more substantial [12].

On the other hand, from the change trend of physical cleaning between cycles, it can be found that the recovery degree of cleaning can be significantly reduced after increasing the concentration of the extract, and the recovery rate of membrane flux is 95% at a concentration of 0.5 M. When the concentration is 1 M, the recovery rate of membrane flux is 90%. When the concentration is 1.5 M, it is reduced to 85%. According to the experimental results, it can also be found that chemical cleaning is required when the reactor runs for three cycles under the condition of a high concentration of the draw solution, and it can be confirmed that the corresponding membrane fouling is faster and the membrane flux decreases more significantly. In this process, the initial flux has a gradual declining trend during this cycle [13]. From this, it can be confirmed that the physical cleaning method has certain limitations, which cannot completely eliminate the pollution layer, and the pollutants continue to accumulate. Under certain conditions, chemical cleaning must be carried out to meet the relevant requirements for continued operation. A 1% NaClO solution was used for chemical cleaning. Three cleanings were conducted in the 45-day operation cycle, of which 36 days entailed chemical cleaning. Figure 2 below shows the change trend of cleaning recovery flux [14].

### 3.2. Characterization of Anolyte Components

Compared with the traditional MFC, the influencing factors of OsMFC membrane fouling are more complex [15] and are related to membrane materials, operating conditions and other factors. This section studies the properties of the mixed solution under operating conditions with different concentrations of the extracted solution. Figure 3 specifically reflects the change rule of the anolyte conductivity with time as the concentration of the draw solution changes. It is found that the anolyte conductivity increases continuously at each concentration with increasing operation time, but the corresponding increase is different. The conductivity of the anode solution increases with the increase in the concentration of the draw solution, which is related to the reverse salt flux characteristics of the FO film. When the concentration of the draw solution is 0.5 M and 1 M, the conductivity of the anode solution shows a slow upward trend [16]. When the concentration of the draw solution is 1.5 M, the conductivity increases continuously in the early and middle stages of the operation but changes slowly in the late stages. This is mainly because the initial flux of the corresponding membrane is low under the condition of a low concentration of the draw solution, which makes it difficult for pollution to form, and the salt reverse osmosis is weak, so the salt accumulation speed is slow. Under the influence of this factor, the increase in mixed-liquid conductivity is small. After increasing the concentration of the extracted solution, the corresponding initial membrane flux also increases to varying degrees, and the salt reverse osmosis is more substantial, which leads to the continuous and rapid accumulation of salt in the reactor and improves the conductivity. After 6 days, the salt reverse osmosis decreases to a certain extent, making the conductivity increase small and stable [17].

Figure 3 specifically reflects the change rule of the anode liquid conductivity with time when the concentration of the absorbing solution changes. It is found that the anode liquid conductivity increases continuously at each concentration with increasing operation time, but the corresponding increase is different. The conductivity of the anode solution increases with the increase in the concentration of the absorbing solution, which is related to the reverse salt flux characteristics of the FO film. When the concentration of the draw solution is 0.5 M and 1 M, the conductivity of the anode solution shows a slow upward trend [18]. When the concentration of the draw solution is 1.5 M, the conductivity increases continuously in the early and middle stages of the operation but changes slowly in the late stages. This is mainly because the initial flux of the corresponding membrane is low under the condition of a low concentration of the draw solution, which makes it difficult for pollution to form, and the salt reverse osmosis is weak, so the salt accumulation speed is slow. Under the influence of this factor, the increase in mixed-liquid conductivity is small. After increasing the concentration of the extracted solution, the corresponding initial membrane flux also increases to varying degrees, and the salt reverse osmosis is more substantial, which leads to the continuous and rapid accumulation of salt in the reactor and improves the conductivity. After 6 days, the salt reverse osmosis decreases to a certain extent, making the conductivity increase small and stable [19].

Figure 4 shows the concentration changes of the anolyte SMP, LB-EPS and TB-EPS during the long-term operation of OsMFC. It can be seen from this that during the operation of the device, the three increase to a certain extent, but the increase is different. Specifically, in the early period of operation (0–15 d), the increase in the three is not obvious. In the middle period of operation (20 days), the specific analysis of the change trend of electrical conductivity shows that the salt concentration keeps increasing, which inhibits microbial activity, and the corresponding EPS and SMP continue to accumulate; the SMP concentration accumulates the most. It can be concluded that the accumulation of SMP and EPS will have a negative impact on the membrane performance [20].

By testing the protein and polysaccharide contents on the membrane surface, it was also found that the polysaccharide content that pollutes the membrane surface was much higher than the protein and was the main pollution component. It can be seen in Figure 5 that the concentrations of proteins and polysaccharides in the anode solution increase due to their accumulation in the anode chamber, but the increases are different. When the extraction solution is 0.5 M, the protein concentration increases by 50% and 200%, and the polysaccharide increases by 60% and 160% when the concentration is controlled at 1 M and 1.5 M. When the concentration reaches 1.5 M, the concentration of these pollutants increases significantly. It can be seen from the analysis that the reasons for the increase in organic matter concentration in the anolyte during this process are as follows: (1) With the increase in the concentration of the draw solution, the membrane flux increases [18]. Under the influence of this factor, the utilization level of organic matter decreases, and the organic matter index of the anode increases. (2) Salt reverse osmosis is more substantial, and the corresponding salt concentration increases, which inhibits microbial activity and its normal growth and metabolism. The specific analysis of the above results also shows that the concentration of proteins is relatively low, and the polysaccharide content is relatively high at the concentration of each extract.

### 3.3. Effect of Membrane Fouling on Power Generation Performance

OsMFC runs the polluted membrane and new membrane alternately during operation and measures the data related to water flux and the current density. The results are shown in Figure 6a. It can be seen from the analysis that the initial water flux of the new membrane is significantly higher than that of the polluted membrane in a given period, about 3.5 times the latter. The water flux of the polluted membrane disappears in a short time, which can be used to confirm that its surface pollution layer is very large. However, after a long time of operation, the new membrane will also produce some pollution, which will lead to a reduction in water flux, which is also related to the concentration polarization factor. The water flux will cause an osmotic dilution effect, and the water flux will decrease greatly under the influence of concentration polarization [21].

As shown in Figure 6b–d, although the polluted membrane water flux is small, the current density is much higher than that of the new membrane, which is much different from that of the traditional MFC. The specific performance result is that the power generation efficiency of the traditional MFC is greatly reduced after serious pollution. According to the analysis of the impedance spectrum and polarization curve, the power density after four operation cycles is higher than that of two and three operation cycles, but they are all larger than that of the new membrane. At the same time, compared with the new membrane, the internal resistance of the polluted membrane is reduced by 40%. According to the analysis of membrane impedance, the decrease in membrane impedance may be due to the increase in reverse salt flux and the increase in the diffusion of positively charged ions. In order to study these two points, it is necessary to study the membrane mass transfer model under membrane fouling conditions [22].

### 3.4. Mass Transfer of Membrane Fouling Model

When the active layer faces the feed liquid, the driving force of the apparent concentration in the FO process includes four parts: (1) the driving force of the effective concentration, Δ ceff, (2) ICP-related driving force loss, Δ C_dicp_, (3) ECP-related losses on the feed solution side, Δc_cecp_, and (4) ECP-related losses on the draw side (Δ cdecp). If the influence of membrane fouling factors is not included in this study, the resistance of water flux and reverse salt flux is only due to the membrane itself, which can be calculated by the following formula [23]:(4)$$J_{w}=\frac{(\pi_{D,\;b}-\pi_{F,\;b})-F_{cecp,\;f}(\pi_{F,\;b}+\frac{J_{s,\;f}}{J_{w,\;f}}\beta RT)-F_{dicp,\;f}\left(\pi_{D,\;b}+\frac{J_{s,\;f}}{J_{w,\;f}}\beta RT\right)}{\mu \left(R_{m,\;w}+R_{f,\;w}\right)}$$
(5)$$J_{s}=\frac{\left(C_{D,\;b}-C_{F,\;b}\right)-F_{cecp,\;f}(C_{F,\;b}+\frac{J_{s,\;f}}{J_{w,\;f}})-F_{dicp,\;f}\left(C_{D,\;b}+\frac{J_{s,\;f}}{J_{w,\;f}}\right)}{R_{m,\;s}+R_{f,\;s}}$$
where μ is the viscosity of the raw solution, m^−1^. The relationship between R_m,w_ and coefficient A is A = 1/(μR_mw_), and R_m,s_ is the salt penetration resistance of the membrane, s⋅m^−1^; the relationship between R_m,s_ and the membrane salt permeability coefficient B is B = 1/R_m,s_. R_f,w_ and R_f,s_ are the mass transfer resistance of water flux and reverse salt flux caused by membrane pollution, and the units are m^−1^ and s⋅m^−1^, respectively; F_cecp,f_ and F_dicp,f_ are the concentration polarization factors of the concentrated external concentration and the diluted internal concentration after membrane pollution and can be determined in the following way during calculation and analysis [21]:

where k_cecp,f_ is the total mass transfer coefficient of the feed solution and the contaminated layer. Since the active layer of the FO membrane almost intercepts all pollutants, k_dicp,f_ = K_dicp_ = D/S (D is the diffusion coefficient of salt in water, and S is the structure coefficient).

### 3.5. Calculation of Solute Mass Transfer Resistance

The solute mass transfer resistance under membrane fouling conditions can be divided into membrane mass transfer resistance (R_ms_) and the pollution-induced mass transfer resistance increment (R_f,s_), where the latter is the reciprocal of (k_f,s_), and its expression is determined on the basis of a certain theoretical derivation [22]:(6)$$R_{f,\;s}=\frac{1}{k_{f,\;s}}=\frac{\delta_{f}}{D^{f}}=\frac{S_{f}}{D}\;$$
where *δ_f_* corresponds to the thickness of the contaminated layer, m; *D^f^* specifically refers to the effective diffusion coefficient, m^2^·s^−1^, which is *D^f^* = *D*⋅*ε_f_*/τ*_f_*, where D is the diffusion coefficient in pure water (m^2^·s^−1^), *ε_f_* is the porosity of the contaminated layer (%), and τ*_f_* is tortuosity. *S_f_* is the structural parameter of the pollution layer, which can be deduced as follows:(7)$$S_{f}=\frac{\delta_{f}\tau_{f}}{\varepsilon_{f}}\;$$

The increase in membrane mass transfer resistance caused by membrane fouling mainly depends on the structural parameter S_f_ of the fouling layer, and S_f_, according to its calculation formula, depends on the thickness of the fouling layer δ_f_, porosity ε_f_ and tortuosity τ_f_. Assuming that the porosity and curvature of the contaminated layer do not change, S_f_ is simply related to the thickness of the contaminated layer δ_f_, where the latter has a positive correlation with the pollutant mass (md) and the corresponding expression [22]:(8)$$\delta_{f}=\frac{m_{d}}{\rho_{f}\left(1-\varepsilon_{f}\right)}$$
(9)$$m_{d}=C_{md}V_{p}C_{f}=C_{md}\overset{t}{\underset{0}{\int }}J_{w,\;f}C_{f}d_{t}$$
where *C**_f_* is the pollutant concentration of the feed liquid, g·m^−3^; ρ*_f_* is the density of the contaminated layer. On this basis, the analysis can determine the solute mass transfer resistance formula as the time changes:(10)$$R_{f,\;s}=\frac{\delta_{f}\tau_{f}}{D_{\varepsilon_{f}}}=\frac{\tau_{f}C_{md}}{\rho_{f}\varepsilon_{f}D\left(1-\varepsilon_{f}\right)}\overset{t}{\underset{0}{\int }}J_{w,\;f}C_{f}dt$$

When the pollutant concentration *C_f_* of the feed solution is controlled at a constant value:(11)$$R_{f,\;s}=\frac{\tau_{f}C_{md}c_{f}}{\rho_{f}\varepsilon_{f}D\left(1-\varepsilon_{f}\right)}\overset{t}{\underset{0}{\int }}J_{w,\;f}dt=C_{fs}\overset{t}{\underset{0}{\int }}J_{w,\;f}dt\;$$
where *C_fs_* is the proportional coefficient of solute mass transfer resistance, s·m^−2^.

### 3.6. Hydraulic Resistance Calculation

The specific analysis of membrane fouling shows that membrane hydraulic resistance can be divided into two parts, namely, hydraulic resistance *R_m,w_* and the corresponding resistance increment *R_f,w_*, which can be expressed as [22]
(12)$$R_{f,\;w}=\frac{180\left(1-\varepsilon_{f}\right)}{\rho_{f}d_{f}^{2}\varepsilon_{f}^{3}}m_{d}$$
where df is the diameter of pollutant particles. Assuming that cf remains unchanged, we can obtain
(13)$$R_{f,\;w}=\frac{180\left(1-\varepsilon_{f}\right)C_{md}c_{f}}{\rho_{f}d_{f}^{2}\varepsilon_{f}^{3}}\overset{t}{\underset{0}{\int }}J_{w,\;f}dt=C_{fw}\overset{t}{\underset{0}{\int }}J_{w,\;f}dt$$
wherein *C_fw_* is the proportional coefficient of the corresponding hydraulic resistance, m^−2^.

### 3.7. Calculation of Solute Concentration on Membrane Surface

By calculating the solute concentration on the membrane surface of the feed solution and the draw solution side when the membrane is polluted, based on the analysis of the relevant experimental parameters, the change in the effective driving force under pollution conditions was determined. The polarization of the external concentration difference will obviously affect the concentration of the solute near the membrane in the feed solution. External concentration polarization after membrane pollution shows very complex characteristics, involving concentration polarization caused by the original boundary layer and concentration polarization caused by ion diffusion being blocked by the pollution layer. Under the influence of these factors, the performance of the membrane is significantly affected. The analysis after membrane fouling shows that the total mass transfer coefficient (*k_cecp,f_*) on the liquid side of the feed can be divided into two parts—the mass transfer coefficient (*k_f,s_*) of the contaminated layer and the boundary layer mass transfer coefficient (*k_cecp,*0*_*) of the contaminated layer surface. Then, *k_cecp,f_* can be calculated by the following formula [23]:(14)$$\frac{1}{k_{cecp,\;f}}=\frac{1}{k_{cecp,\;0}}+\frac{1}{k_{f,\;s}}=\frac{1}{k_{cecp}}+R_{f,\;s}$$

The concentration polarization factor *F_cecp,f_* and solute concentration on the membrane surface after membrane pollution are determined by the following formula analysis:(15)$$C_{F,\;m}=C_{F,\;b}+F_{cecp,\;f\;}(C_{F,\;b}+\frac{J_{s,\;f}}{J_{w,\;f}})$$

In AL-FS mode, membrane fouling will not significantly affect the support layer, so the solute concentration on the membrane surface on the liquid side can be calculated as follows:(16)$$C_{D,\;m}=C_{D,\;b}-F_{dicp,\;f\;}\left(C_{D,\;b}+\frac{J_{s,\;f}}{J_{w,\;f}}\right)$$

### 3.8. Simulation of Solution Concentration on Both Sides of Membrane under Continuous-Flow OsMFC Operation

To simulate the performance change of the system under membrane fouling conditions, changes in the solution concentration and volume on both sides of the FO membrane with time were studied [24].
(17)$$\frac{d_{V_{F}}}{d_{t}}=Q_{in,\;F}-J_{w,\;f}A_{m}-Q_{out,\;F}$$
(18)$$\frac{d_{V_{D}}}{d_{t}}=Q_{in,\;D}+J_{w,\;f}A_{m}-Q_{out,\;D}$$

In the formulas, *V_F_* and *V_D_* are the volume of the feed and draw solutions, respectively, and *Q_in_* and *Q_out_* are the inlet and outlet flows, respectively.

### 3.9. Model Calculation and Analysis

In the process of analyzing the infiltration resistance parameters of FO membrane fouling mass transfer, two key parameters need to be determined first, namely, the salt mass transfer resistance coefficient *C_fs_* and the hydraulic resistance coefficient *C_fw_*. During the measurement of membrane fouling,the corresponding water flux and reverse salt flux were determined by adding the mass of deionized water and the conductivity of the solution on the feed liquid side.The collected experimental data were fitted,mainly to determine the reference values of *C_fw_* and *C_fs_* by least-square fitting,calculate the dynamic changes in reverse salt flux resistance and driving force during membrane fouling, and analyze the effects of different membrane fouling conditions shown in Table 1 [25].

The salt mass transfer resistance coefficient and hydraulic resistance coefficient introduced in the model were obtained through nonlinear least-square regression (Figure 7). The hydraulic resistance coefficient is 5.29 × 10^15^. The comparative analysis shows that this resistance is significantly higher than the material’s own resistance, which is 22.4 times the latter. The increased salt mass transfer resistance coefficient calculated under the same pollution conditions is 1.41 × 10^6^, 0.5 times that of the membrane itself. It can be concluded that membrane fouling will have a more obvious impact on hydraulic resistance under such experimental conditions. During the experimental study, the concentration on both sides of the membrane was set to remain unchanged, and the volume of the feed solution was controlled at a higher level so that the salt accumulation would not significantly affect the osmotic pressure difference on both sides. Under this condition, the decrease in FO water flux is mainly related to membrane pollution [26]. During the decline, the reverse salt flux continues to increase. At the initial stage of membrane fouling, i.e., 0–8 h, water flux decreases rapidly, and the reverse salt flux increases rapidly. Later, when the water flux decreases slowly, the reverse salt flux also slows down. Due to the increase in FO reverse salt flux under membrane fouling, the salt concentration in the feed solution is higher than that in the blank control group [27].

### 3.10. Concentration Polarization Factor and Effective Concentration Difference

In order to analyze the phenomenon that membrane fouling accelerates desalination flux, a model was established to calculate the driving force and resistance of desalination flux in the process of membrane fouling. The main factor affecting the operation effect is the internal resistance power loss caused by high internal resistance, which mainly includes the ohmic internal resistance loss, activation internal resistance loss and concentration difference loss [28]. Research has shown that replacing CEM with FO or PEM can affect the membrane impedance in ohmic internal resistance. However, as a characteristic of OsMFC, how the water flux affects its power generation capacity deserves further study [29]. Previous studies have confirmed that OsMFC promotes ion transport between cathode and anode chambers due to the generation of water flux, which indicates that the FO membrane as an MFC separation material has a lower blocking effect than CEM and AEM membranes. At the same time,water flux can also promote proton transfer and ease the decrease in anode pH and the increase in cathode pH [30].The research results also show that OsMFC can stabilize the system pH and reduce the system overvoltage [31]. It can be seen in Figure 8a that the external concentration polarization factor (*F_cecp,f_*) is increased from 0.6 to 1.23. Analysis has shown that the reason for the change in cooperation is mainly caused by concentration polarization after pollution [32]. In this process, the internal concentration polarization factor (*F_dicp,f_*) decreases from 0.8 to 0.52 due to the decrease in water flux. Although the changes in *F_dicp_* are not obvious, the salt concentration index on the membrane surface on the draw solution side increases significantly after their changes. In addition, a comparative analysis shows that during the period of the water flux decline, the increase rate of *C_F,m_* concentration is increased [33]. During this change, *C_F,m_* only increased from 0.08 M to 0.19 M. This also shows that after membrane fouling, the effective concentration difference on both sides increases, which promotes the desalination process, and the corresponding flux increases. The internal concentration polarization will more notably affect the driving force loss of the FO process. After pollution, the internal concentration polarization factor will significantly decrease. Under the cumulative effect, the membrane surface concentration will increase, and the reverse salt flux will be enhanced. This phenomenon can be regarded as a special self-compensation effect of this kind of membrane [34].

From the calculation of hydraulic resistance and reverse salt resistance coefficients, it can be seen that both the hydraulic resistance and reverse salt resistance of the membrane increase in the process of membrane pollution, but the amount of the increase is different; that is, the influence of membrane pollution on hydraulic resistance is stronger than that on reverse salt resistance, as shown in Figure 8b. This is related to the nature of pollutants. The pollutants form a dense gel layer on the membrane surface, which significantly increases the hydraulic resistance, while the gel layer has a certain ion-exchange capacity; therefore, the increase in inorganic salt ion penetration resistance is small [35].

### 3.11. Hydraulic Resistance and Reverse Salt Resistance

At the same time, the driving force changes of water flux and reverse salt flux before and after membrane pollution were compared. After processing the relevant data, Figure 9 was drawn. In the model analysis, it was assumed that the osmotic pressure and the concentration of the salt solution followed the Van’t Hoff model [36], so the driving force changes of water and salt penetration were consistent. Upon the completion of the membrane pollution experiment, both of them had increased by 2.3 times. The hydraulic resistance of the FO membrane increased to 4.97 times the initial value, while the salt penetration resistance only increased to 1.09 times the initial value. Under the influence of this difference, membrane pollution makes the change trends of water and reverse salt flux reverse; specifically, the former continuously decreases during the aggravation of pollution and then decreases to 45% of the original. However, during the process of increasing membrane pollution, the reverse salt flux increased to a certain extent and was 2.3 times the original value at the end [36,37]. According to the above results, it can be inferred that FO membrane fouling increases reverse salt flux, mainly because membrane fouling causes concentration polarization in the FO membrane, which enhances the reverse salt trend, and the enhancement effect is much higher than the mass transfer resistance. Membrane pollution accelerates the reverse salt process [38].

## 4. Conclusions

In this study, a mass transfer model of a contaminated membrane was innovatively constructed. Through the analysis of the hydraulic resistance coefficient and the salt mass transfer resistance coefficient, the mass transfer law of the contaminated membrane was studied. The following conclusions were drawn.

The hydraulic resistance coefficient of the contaminated membrane increased to 4.97 times the initial value, while the salt mass transfer resistance coefficient did not change significantly. Under the influence of this difference, membrane fouling made the flux of the two completely opposite. The specific performance result is that the former continuously decreased during the process of pollution intensification and then decreased to 45% of the original. However, with increasing membrane fouling, the reverse salt flux increased to a certain extent, which was 2.3 times the original value at the end. According to the above results, it can be inferred that FO membrane pollution increases the reverse salt flux mainly because the membrane pollution causes concentration polarization in the FO membrane, which enhances the reverse salt trend; the enhancement effect is much higher than the mass transfer resistance, and the membrane pollution accelerates the reverse salt process.

The fouling behavior of the FO membrane during the long-term operation of OsMFC was analyzed. Since the salt mass transfer resistance system is not significantly increased by membrane pollution after membrane pollution, the driving force of the reverse salt flux is enhanced in a disguised way, resulting in the reverse salt flux increasing to 2.3 times that of the new membrane, which is far higher than the impact of the water flux decline caused by membrane pollution, thus increasing the concentration on the feed liquid side, reducing the membrane resistance and overall impedance, and promoting the improvement of the power generation efficiency.

Through the model simulation of the concentration inside the membrane, the change in the membrane thickness direction and the phase transition of the internal structure of the membrane from the dry state to the expanded state were analyzed, which were influenced by water flux, thus further explaining the important role of the microenvironment in the membrane in reducing membrane impedance.

## Figures and Tables

**Figure 1 membranes-12-01165-f001:**
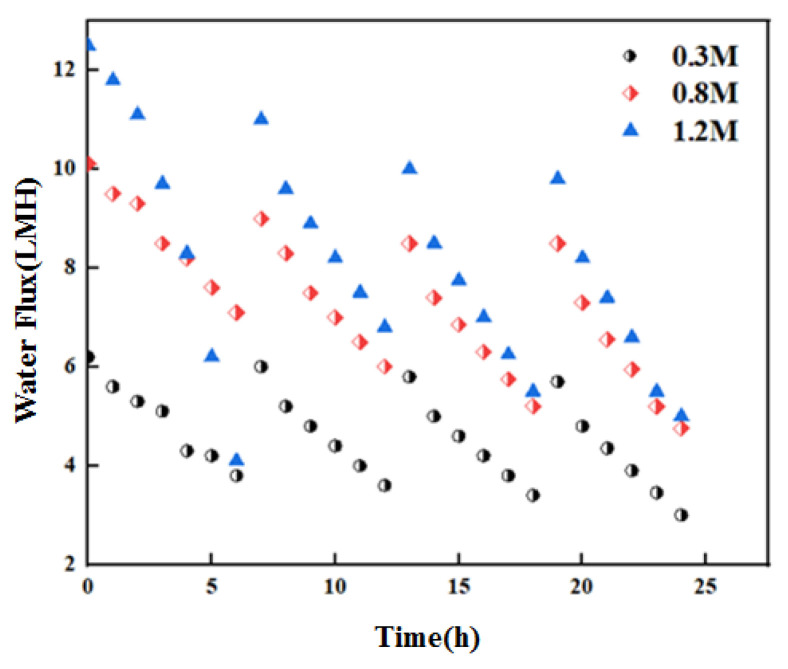
Change in membrane flux during operation of OsMFC.

**Figure 2 membranes-12-01165-f002:**
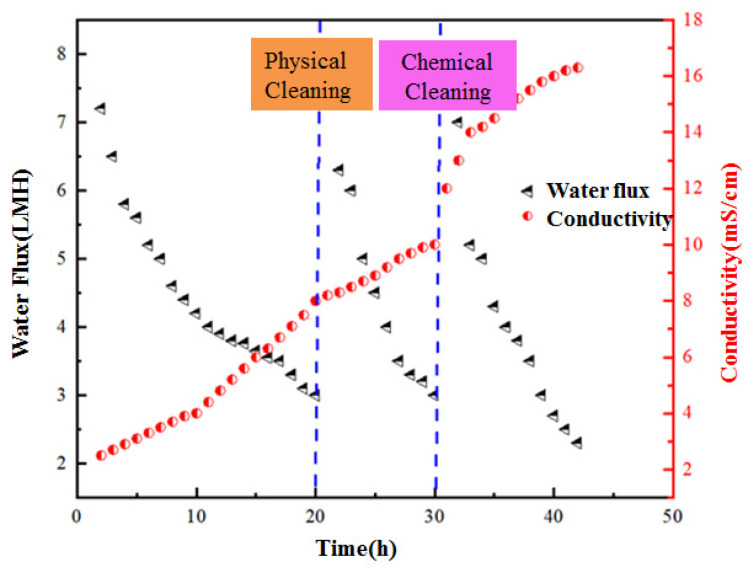
OsMFC water flux and conductivity change with operation time.

**Figure 3 membranes-12-01165-f003:**
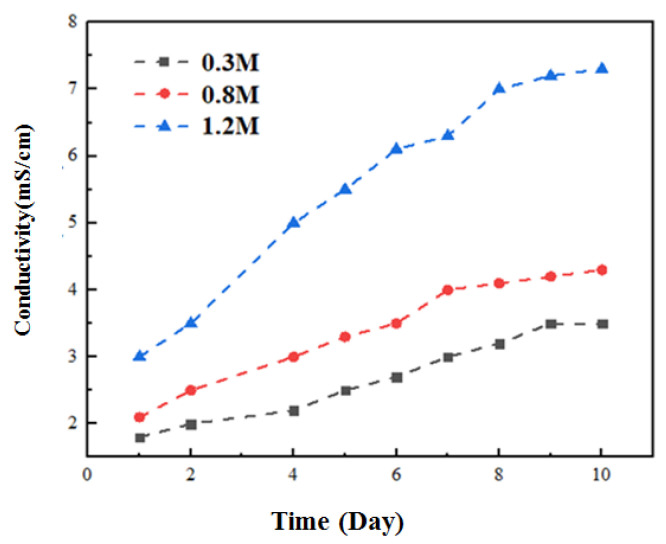
Change in anolyte conductivity during operation of OsMFC.

**Figure 4 membranes-12-01165-f004:**
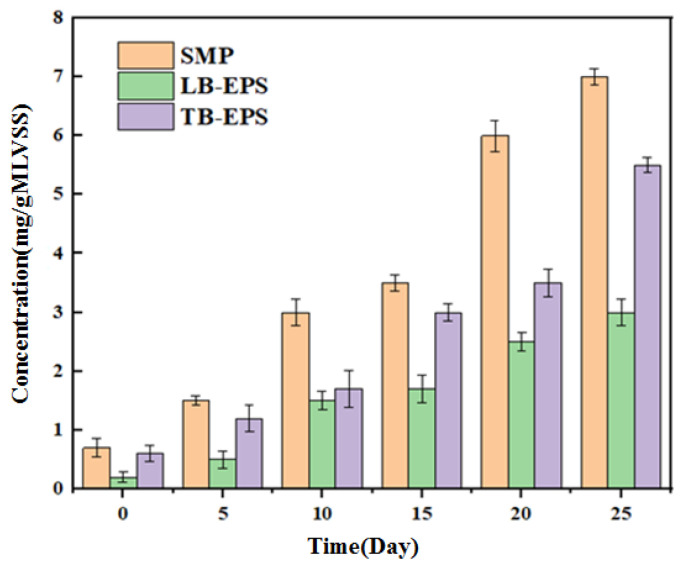
Changes in microbial metabolite concentration and conductivity in OsMFC anolyte with operation time.

**Figure 5 membranes-12-01165-f005:**
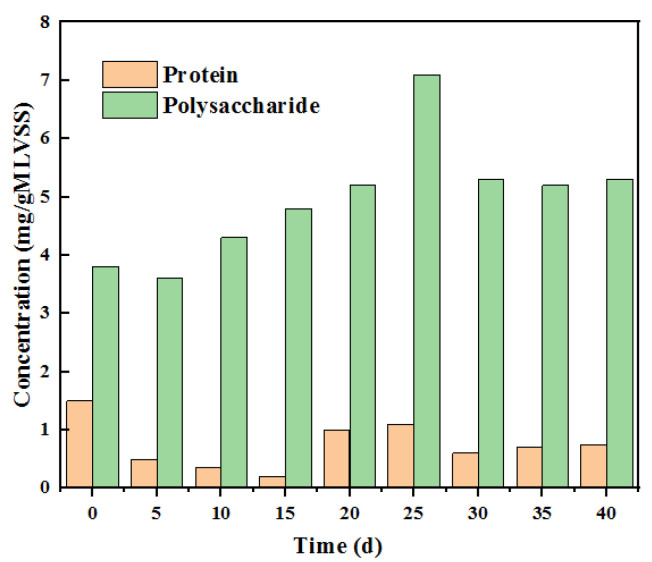
Changes in membrane surface polysaccharides and proteins with time during the operation of OsMFC.

**Figure 6 membranes-12-01165-f006:**
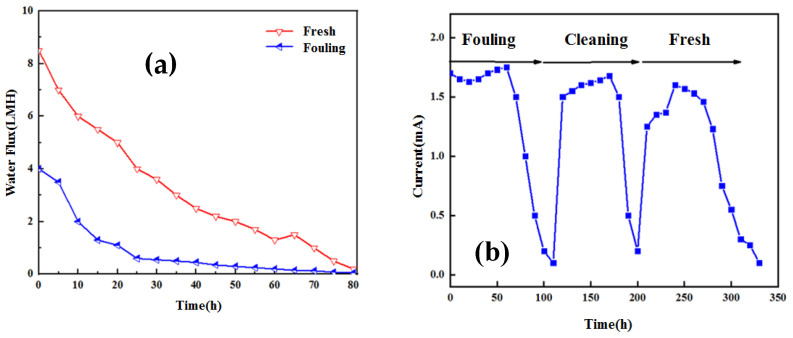
Testing of new membranes and pollutants: water flux change of new membranes and polluted membranes (**a**); current curve of new membrane and polluted membrane (**b**); Nyquist diagram of new and old membranes (**c**); OsMFC polarization curves (**d**) of new membranes and membranes with different operating cycles.

**Figure 7 membranes-12-01165-f007:**
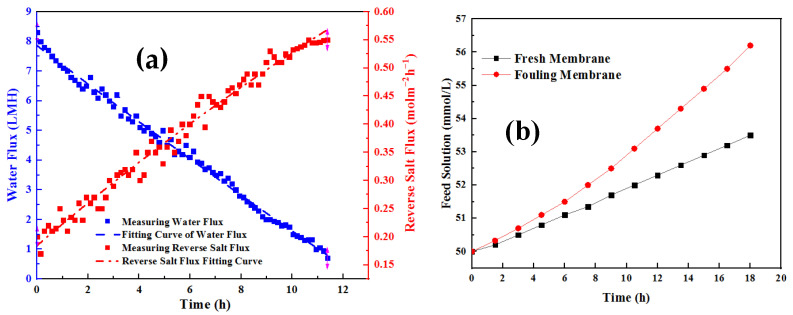
Experimental and simulation results of water flux and reverse salt flux during membrane fouling test (**a**) and the change trend of salt concentration in feed liquid with and without pollution calculated by the model (**b**).

**Figure 8 membranes-12-01165-f008:**
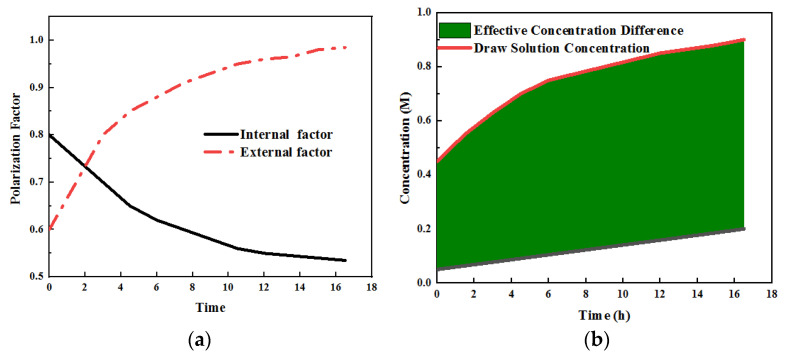
(**a**) The polarization factors of internal and external concentration differences calculated by the model, as well as (**b**) the salt concentration and effective concentration difference on both sides of the active layer of FO membrane during membrane fouling (M).

**Figure 9 membranes-12-01165-f009:**
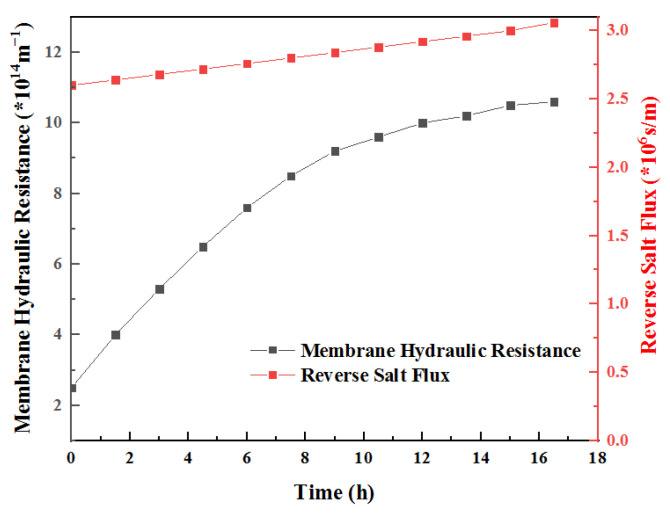
Changes in hydraulic resistance and reverse salt resistance of FO membrane during membrane fouling.

**Table 1 membranes-12-01165-t001:** Experimental conditions and model parameters.

Parameter	Symbol	Value	Unit	Reference
Conditions and Parameters				
Initial concentration of feed solution	*C_F, 0_*	0.035	mol·L^−1^	[25]
Inlet salt concentration	*C_in, F_*	0	mol·L^−1^	[25]
Volume of feed solution	*V_F_*	3.5	L	[23]
Draw solution concentration	*C_D, b_*	1.2	mol·L^−1^	[25]
Effective area of membrane	*A_m_*	3.1 × 10^−3^	m^2^	[26]
Temperature	*T*	298.15	K	[26]
Viscosity of pure water	*μ*	9.9 × 10^−4^	Pa.s	[23]
Diffusion coefficient of salt in water	*D*	1.35 × 10^−9^	m^2^.s^−1^	[23]
Membrane property parameters				
Hydraulic resistance	*R_mw_*	2.39 × 10^14^	m^−1^	[22]
Salt mass transfer resistance	*R_ms_*	2.85 × 10^6^	s·m^−1^	[22]
Structural parameters	*S*	425	μm	[23]
Boundary layer thickness	*δ*	125	μm	[25]
Boundary layer mass transfer coefficient	*K_cecp_*	1.3 × 10^−5^	m·s^−1^	Calculation
Mass transfer coefficient of support layer	*K_dicp_*	3.15 × 10^−6^	m/s	Calculation
Pollution layer property parameters				
Salt mass transfer resistance coefficient	*C_fs_*	1.51 ± 0.07	10^16^ s·m^−2^	Correcting
Hydraulic resistance coefficient	*C_fw_*	5.39 ± 0.1	10^15^ m^−2^	Correcting

## Data Availability

The data presented in this study are available on request from the corresponding author.

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
