# Peer review of "Influence of Membrane Fouling and Reverse Salt Flux on Membrane Impedance of Forward Osmosis Microbial Fuel Cell"

_membranes, 2022, doi:10.3390/membranes12111165_

Round 1

Reviewer 1 Report

In general, the article is tackling a very important and interesting topic, discussing the membrane fouling and reverse salt flux on membrane impedance, here are some suggestions:

1)    The authors should add to the introduction an explicit description of the influence of membrane fouling type and membrane fouling on the bioreactor

2)    The sentence in lines 210-212 requires clarification and reference to the source.

3)    In this paper, you mentioned that the FO membrane is different from the traditional membrane in that it can continue to generate electricity after membrane pollution. How about the effect of electricity generation compared with the new membrane.

4)    Check the Formula 2 in line 98.

Reviewer 2 Report

This paper has done important work in studying the the mechanism of continuous power generation after membrane fouling is analyzed through the model. Here are some suggestions for your reference

 1)    The authors should add to the introduction an explicit description of the novelty of the work.

2)    What is the direct impact of membrane fouling on water flux and salt reverse flux?

3)    The authors need to describe more clearly what are the main types of membrane pollution?

4)    The sentence in lines 317-319 requires clarification and reference to the source.
